# Configuration and Intercomparison of Deep Learning Neural Models for Statistical Downscaling

Jorge Baño-Medina [1], Rodrigo Manzanas [2], and José Manuel Gutiérrez [1]

[1]Santander Meteorology Group, Institute of Physics of Cantabria (CSIC-Univ. of Cantabria), Santander (Spain)
[2]Santander Meteorology Group, Dpto. de Matemática Aplicada y Ciencias de la Computación, Universidad de Cantabria, Santander (Spain)

**Correspondence:** Jorge Baño–Medina (bmedina@ifca.unican.es)

**Abstract.** Deep learning techniques (in particular convolutional neural networks, CNNs) have recently emerged as a promising approach for statistical downscaling due to their ability to learn spatial features from huge spatio-temporal datasets. However, existing studies are based on complex models, applied to particular case studies and using simple validation frameworks, which makes difficult a proper assessment of the (possible) added value offered by these techniques. As a result, these models are usually seen as black-boxes generating distrust among the climate community, particularly in climate change applications.

In this paper we undertake a comprehensive assessment of deep learning techniques for continental-scale statistical downscaling, building on the VALUE validation framework. In particular, different CNN models of increasing complexity are applied for downscaling temperature and precipitation over Europe, comparing them with a few standard benchmark methods from VALUE (linear and generalized linear models) which have been traditionally used for this purpose. Besides analyzing the adequacy of different components and topologies, we also focus on their extrapolation capability, a critical point for their potential application in climate change studies. To do this, we use a warm test period as surrogate of possible future climate conditions. Our results show that, whilst the added value of CNNs is mostly limited to the reproduction of extremes for temperature, these techniques do outperform the classic ones for the case of precipitation for most aspects considered. This overall good performance, together with the fact that they can be suitably applied to large regions (e.g. continents) without worrying about the spatial features being considered as predictors, can foster the use of statistical approaches in international initiatives such as CORDEX.

## 1 Introduction

The coarse spatial resolution and systematic biases of Global Climate Models (GCMs) are two major limitations for the direct use of their outputs in many sectoral applications, such as hydrology, agriculture, energy or health, particularly for climate change impact studies (SD, Maraun and Widmann, 2017). These applications typically involve the use of sectoral models (e.g. crop or hydrological models) and/or climate indices (e.g. frost days or warm spells) which require regional to local weather (daily) series of different variables (precipitation, temperature, radiation, wind, etc.) over multiple decades representative of the historical and future climates (see, e.g. Galmarini et al., 2019; Ba et al., 2018; Sanderson et al., 2017; Teutschbein et al., 2011; Wang et al., 2017). Moreover, the results of these studies are sensitive to different aspects of the climate data, such as the

temporal structure (e.g. in agriculture or energy), the spatial and/or inter-variable structure (e.g. in hydrology), or the extremes (e.g. in hydrology and health).

In order to bridge this gap, different *statistical downscaling* (SD, Maraun and Widmann, 2017) methods have been developed building on empirical relationships established between informative large-scale atmospheric variables (predictors) and local/regional variables of interest (predictands). Under the perfect prognosis approach, these relationships are learned from (daily) data using simultaneous observations for both the predictors (from a reanalysis) and predictands (historical local or gridded observations), and are subsequently applied to GCM simulated predictors (multi-decadal climate change projections under different scenarios), to obtain locally downscaled values (see, e.g., Gutiérrez et al., 2013; Manzanas et al., 2018).

A number of standard perfect prognosis SD (hereafter just SD) techniques have been developed during the last two decades building mainly on (generalized) linear regression and analog techniques (Gutiérrez et al., 2018). These standard approaches are widely used by the downscaling community and several intercomparison studies have been conducted to understand their advantages and limitations taking into account a number of aspects such as temporal structure, extremes, or spatial consistency. In this regard, the VALUE (Maraun et al., 2015) initiative proposed an experimental validation framework for downscaling methods and conducted a comprehensive intercomparison study over Europe with over 50 contributing standard techniques (Gutiérrez et al., 2018).

Besides these standard SD methods, a number of machine learning techniques have been also adapted and applied for downscaling. For instance, the first applications of neural networks date back to the late 90s (Wilby et al., 1998; Schoof and Pryor, 2001). More recently, other alternative machine learning approaches have been applied, such as support vector machines (SVMs, Tripathi et al., 2006), random forests (Pour et al., 2016; He et al., 2016) or genetic programming (Sachindra and Kanae, 2019). There have been also a number of intercomparison studies analyzing standard and machine learning techniques (Wilby et al., 1998; Chen et al., 2010; Yang et al., 2016; Sachindra et al., 2018), with an overall consensus that no technique clearly outperforms the others and that limited added value —in terms of performance, interpretability and parsimony— is obtained with sophisticated machine learning options, particularly in the context of climate change studies.

In the last decade, machine learning has gained a renewed attention in several fields, boosted by major breakthroughs obtained with Deep Learning (DL) models (see Schmidhuber, 2015, for an overview). The advantage of DL resides in its ability to extract high-level feature representations in a hierarchical way due to its (deep) layered-structure. In particular, in spatiotemporal datasets, convolutional neural networks (CNN) have gained great attention due to its ability to learn spatial features from data (LeCun and Bengio, 1995). DL models allow to automatically treat high-dimensional problems avoiding the use of conventional feature extraction techniques (e.g. Principal Components, PCs), which are commonly used in more classic approaches (e.g., linear models and traditional fully-connected neural networks). Moreover, new efficient learning methods (e.g. batch, stochastic, and mini-batch gradient descent), regularization options (e.g. dropout), and computational frameworks (e.g. TensorFlow; see Wang et al., 2019, for an overview) have popularized the use of DL techniques, allowing to efficiently learn convolutional neural networks from (big) data avoiding overfitting. Different configurations of CNNs have proven successful in a variety of problems in several disciplines, particularly in image recognition (Schmidhuber, 2015). There are also a number of recent successful applications in climate science, including the detection of extreme weather events (Liu

et al., 2016), the estimation of cyclone's intensity (Pradhan et al., 2018), the detection of atmospheric rivers (Chapman et al., 2019), the emulation of model parameterizations (Gentine et al., 2018; Rasp et al., 2018; Larraondo et al., 2019) and full simplified models (Scher and Messori, 2019). The reader is referred to Reichstein et al. (2019) for a recent overview.

There have been some attempts to test the application of these techniques for SD, including simple illustrative examples of super-resolution approaches to recover high-resolution (precipitation) fields from low resolution counterparts with promising results (Vandal et al., 2017b; Rodrigues et al., 2018). In the context of perfect prognosis SD, deep learning applications have applied complex convolutional-based topologies (Vandal et al., 2017a; Pan et al., 2019), autoencoder arquitechtures (Vandal et al., 2019) and long-short term memory (LSTM) networks (Misra et al., 2018; Miao et al., 2019) over small case study areas and using simple validation frameworks, resulting in different conclusions about their performance, as compared to other standard approaches. Therefore, these complex (out-of-the-shelve in many cases) models are usually seen as black-boxes generating distrust among the climate community, particularly in climate change problems. Recently, Reichstein et al. (2019) outlined this problem and encouraged research towards the understanding of deep neural networks in climate science.

In this study we aim to shed light on this problem and perform a comprehensive evaluation of deep SD models of increasing complexity, assessing the particular role of the different elements conforming the deep neural network architecture (e.g., convolutional and fully-connected or dense layers). In particular, we use the VALUE validation framework over a continental region (Europe) and compare deep SD methods with a few standard benchmark methods best performing in the VALUE intercomparison (Gutiérrez et al., 2018). Besides this, we also focus on the extrapolation capability of the different methods, which is fundamental for climate change studies. Overall, our results show that simple deep CNNs outperform standard methods (particularly for precipitation) in most of the aspects analyzed.

The code needed to fully replicate the experiments and results shown in this paper is freely available as jupyter notebooks at the DeepDownscaling GitHub repository (https://github.com/SantanderMetGroup/DeepDownscaling; Baño Medina et al., 2020). In addition, in this paper we introduce `downscaleR.keras`, an extension of the `downscaleR` (Bedia et al., 2019) package that integrates `keras` into the `climate4R` (Iturbide et al., 2019) framework (see the *code availability* section).

## 2 Experimental Intercomparison Framework

### 2.1 Area of Study and Data

The VALUE COST Action (2012-2015) developed a framework to validate and intercompare downscaling techniques over Europe, focusing on different aspects such as temporal and spatial structure and extremes (Maraun et al., 2015). The experimental framework for the first experiment (downscaling with 'perfect' reanalysis predictors) is publicly available at http://www.value-cost.eu/validation as well as the intercomparison results for over 50 different standard downscaling methods (Gutiérrez et al., 2018). Therefore, VALUE offers a unique opportunity for a rigorous and comprehensive intercomparison of different deep learning topologies for downscaling.

In particular, VALUE propose the use of twenty standard predictors from the ERA-Interim reanalysis, selected over a European domain (ranging from $36°$ to $72°$ in latitude and from $-10°$ to $32°$ in longitude, with a $2°$ resolution) for the 30-year

period 1979-2008. This predictor set is formed by five large-scale thermodynamic variables (geopotential height, zonal and meridional wind, temperature, and specific humidity) at four different vertical levels (1000, 850, 700 and 500 hPa) each. The left column of Figure 1 shows the climatology (and the grid) of two illustrative predictors used in this study.

The target predictands considered in this work are surface (daily) mean temperature and accumulated precipitation. Instead of the 86 representative local stations used in VALUE, we used the observational gridded dataset from E-OBS v14 ($0.5°$ resolution). Note that this extended experiment allows for a better comparison with dynamical downscaling experiments carried out under the CORDEX initiative (Gutowski Jr. et al., 2016). The right column of Figure 1 shows the climatology of the two target predictands, temperature and precipitation.

Daily standardized predictor values are defined considering the closest ERA-Interim gridboxes (one or four) to each E-OBS gridbox for the benchmarking linear and generalized linear techniques (see Section 2.3). However, the entire domain is used for the deep learning models, which allows to test their suitability to automatically handle high-dimensional input data, extracting relevant spatial features (note that this is particularly important for continental wide applications).

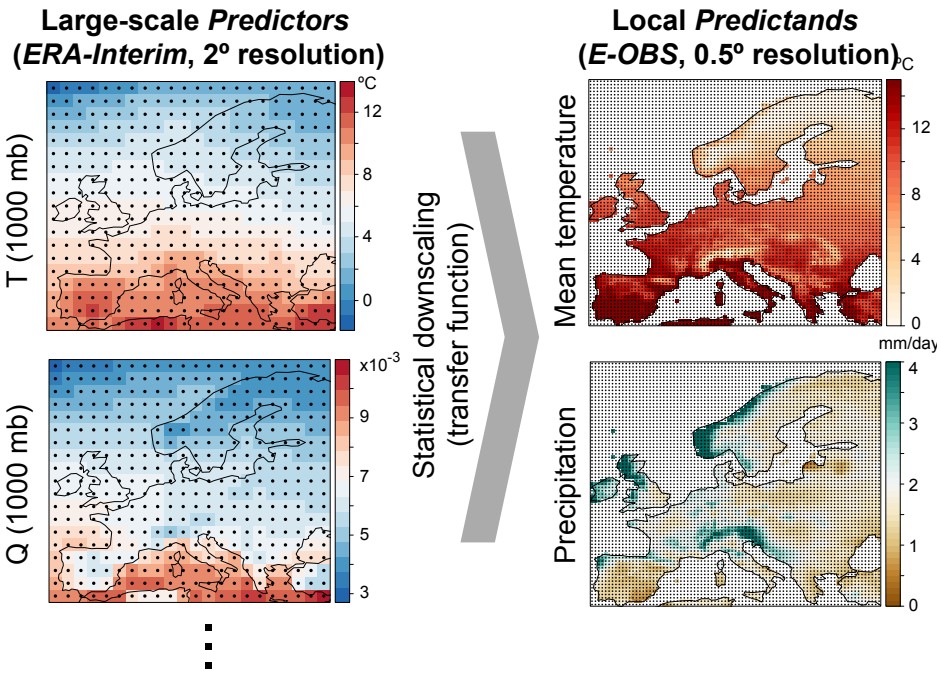

**Figure 1.** Climatology for (left) two typical predictors (air temperature, T, and specific humidity, Q, at 1000 mb), as given by the ERA-Interim reanalysis ($2°$) and (right), the observed target variables of this work, temperature and precipitation from E-OBS ($0.5°$). Dots indicate the center of each gridbox.

## 2.2 Evaluation Indices and Cross-Validation

The validation of downscaling methods is a multi-faceted problem with different aspects involved such as the representation of extremes (Hertig et al., 2019) or the temporal (Maraun et al., 2019) and spatial (Widmann et al., 2019) structure. VALUE developed a comprehensive list of indices and measures (available at the VALUE Validation Portal: http://www.value-cost.eu/validationportal) which allows to properly evaluate most of these aspects. Moreover, an implementation of these indices in an R package (VALUE, https://github.com/SantanderMetGroup/VALUE) is available for research reproducibility. In this work we consider the subset of VALUE metrics shown in Table 1 to assess the performance of the downscaling methods to reproduce the observations. Note that different metrics are considered for temperature and precipitation.

| Description | Variable | Units |
|---|---|---|
| Bias (for the mean) | temp., precip. | $^{\circ}C$ , % |
| Bias (for the 2nd percentile, P2) | temp. | $^{\circ}C$ |
| Bias (for the 98th percentile, P98) | temp., precip. | $^{\circ}C$ , % |
| Root Mean Square Error (RMSE) | temp., precip. | $^{\circ}C, mm/day$ |
| Ratio of standard deviations | temp. | - |
| Pearson correlation | temp. | - |
| Spearman correlation | precip. | - |
| ROC Skill Score (ROCSS) | precip. | - |
| Bias (warm annual max spell, WAMSl) | temp. | days |
| Bias (cold annual max spell, CAMS) | temp. | days |
| Bias (wet annual max spell, WetAMS) | precip. | days |
| Bias (dry annual max spell, DryAMS) | precip. | days |
| Bias (lag 1 autocorrelation, AC1) | temp. | - |
| Bias (relative amplitude of the annual cycle) | precip. | - |

**Table 1.** Subset of VALUE metrics used in this study to validate the different downscaling methods considered (see Table 2). The symbol '-' denotes adimensionality.

For temperature, biases are given as absolute differences (in °C), whereas for precipitation they are expressed as relative differences with respect to the observed value (in %). Note that, beyond the bias in the mean, we also assess the bias in extreme percentiles, in particular the 2nd percentile (P2, for temperature) and the 98th (P98, for both temperature and precipitation). We also compute the biases for four temporal indices used in VALUE: the median warm (WAMS) and cold (CAMS) annual max spells for temperature and the median wet (WetAMS) and dry (DryAMS) annual max spells for precipitation. In addition to the latter temporal metrics we include the (lag 1) autocorrelation (AC1) for temperatures and the annual cycle relative amplitude for precipitation, the latter computed as the difference between maximum and minimum values of the annual cycle (defined using a 30-day moving window over calendar days), relative to the mean of these two values. We also consider the Root Mean Squared

Error (RMSE), which measures the average magnitude of the forecast errors; in the case of precipitation this metric is calculated conditioned to wet observed days (rainfall > 1 mm). To evaluate how close the predictions follow the observations, we also assess correlation, in particular the Pearson coefficient for temperature and the Spearman rank one (adequate for non-gaussian variables) for precipitation; for the particular case of temperature, the seasonal cycle is removed from both observations and predictions in order to avoid its (known) effect on the correlation. This is done by removing the annual cycle defined by a 31-day moving window centered on each calendar day. For this variable we also consider the ratio of standard deviations, i.e., that of the predictions divided by that of the observations. Finally, to evaluate how well the probabilistic predictions of rain occurrence discriminate the binary event rain/no rain, we consider the ROC Skill Score (ROCSS) (see, e.g. Manzanas et al., 2014), which is based on the area under the ROC curve (see Kharin and Zwiers, 2003, for details).

The VALUE framework builds on a cross-validation approach in which the 30-year period of study (1979-2008) is chronologically split into five consecutive folds. We are particularly interested in analyzing the out-of-sample extrapolation capabilities of the deep SD models. Therefore, following the recommendations of Riley (2019, *"the question you want to answer should affect the way you split your data"*), we focus on the last fold, for which warmer conditions have been observed. Therefore, in this work we apply a simplified hold-out approach using the period 2003-2008 for validation, and training the models using the remaining years (1979-2002). Figure 2 shows the climatology of the train period for both temperature and precipitation (top and bottom panel, respectively), as well as the mean differences between the test and the train periods (taken the latter as reference). For temperature, warmer conditions are observed in the test period —over $0.7°$ for both mean values and extremes,— being especially significant for the 2nd percentile (cold days), for which temperatures increase up to $2°$ in northern Europe, compared with the training period. This allows us to estimate the extrapolation capabilities of the different methods, which is particularly relevant for climate change studies.

Importantly, note that the differences between the test and train periods in Figure 2 reveal some inconsistencies in the dataset for both temperature (Southern Iberia and Alps) and precipitation (Northeastern Iberia and the Baltic states). This may be an artifact due to changes or interruptions in the national station networks used to construct E-OBS and may not correspond to a real change in the dataset. This will be taken into account when analyzing the results in Section 4.

## 2.3 Standard Statistical Downscaling Methods used for Benchmarking

We use as benchmark some state-of-the-art standard techniques which ranked among the top in the VALUE intercomparisson experiment. In particular, multiple linear and generalized linear regression models (hereafter referred to as GLM) exhibited good overall performance for temperature and precipitation, respectively (Gutiérrez et al., 2018). Here, we consider the version of these methods described in Bedia et al. (2019) which use the predictor values in the four gridboxes closest to the target location. This choice is a good compromise between feeding the model with full spatial information (all gridboxes, which is problematic due to the resulting high-dimensionality) and insufficient spatial representation when considering a single gridbox. For the sake of completeness we also illustrate the results obtained with a single gridbox, in order to provide an estimate of the added value of extending the spatial information considered for the different variables. These benchmark models are denoted GLM1 and GLM4 for one and four gridboxes, respectively (first two rows in Table 2).

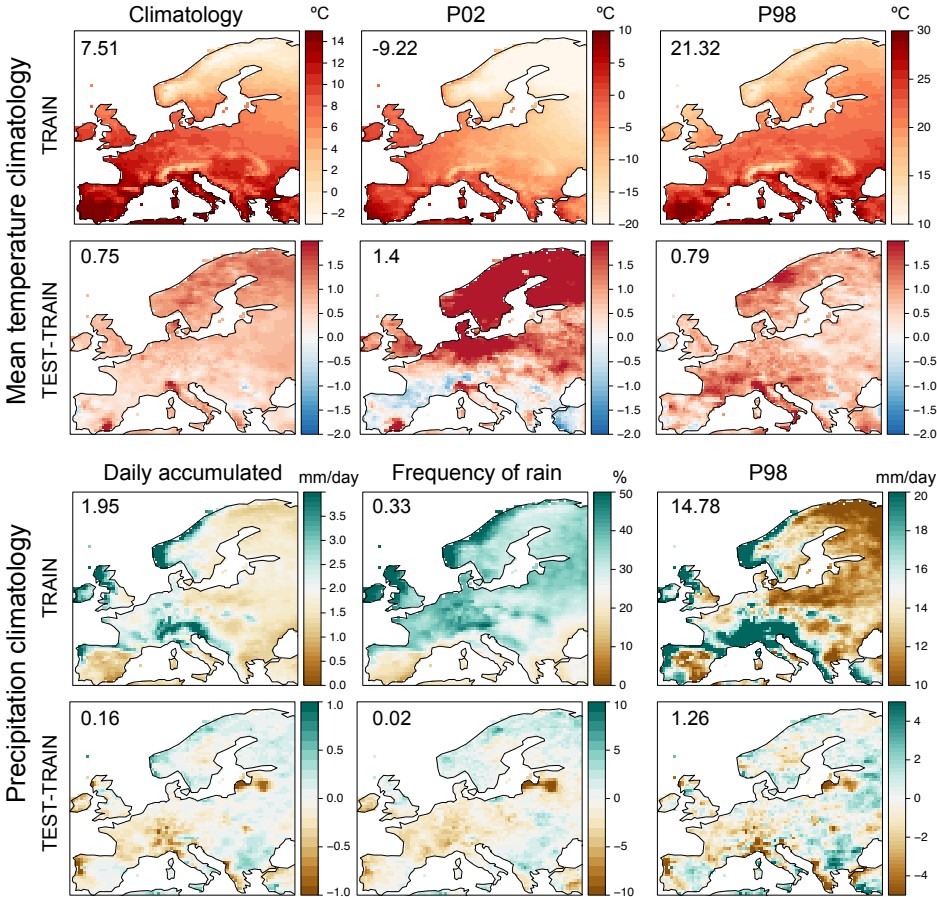

**Figure 2.** Top panel, top row: E-OBS climatology for the mean value, the P02 and the P98 of temperature in the train period (1979-2002). Top panel, bottom row: Mean difference between the test and train periods (the latter taken as reference) for the different quantities shown in the top row. Bottom panel: As the top panel, but for precipitation showing the mean value, the frequency of rainy days and the P98. In all cases, the numbers within the panels indicate the spatial mean values.

In the case of temperature a single multiple regression model (i.e. GLM with Gaussian family) is used, whereas for precipitation two different GLMs are applied, one for the occurrence ($precipitation > 1mm$) and one for the amount of precipitation, using binomial and Gamma families with logarithmic link, respectively (see, e.g., Manzanas et al., 2015). In this case, the values from the two models are multiplied to obtain the final prediction or precipitation, although occurrence and amount are also evaluated separately.

## 3 Deep Convolutional Neural Networks

Despite the success of deep learning in many fields, these complex and highly-nonlinear models are still seen as black boxes generating distrust among the climate community, particularly in climate change problems, since their validation and general-

| Model | Architecture | Rationale |
|---|---|---|
| **GLM1** | 20-1 ($\times$ 3258) | Simplest linear local model for benchmarking |
| **GLM4** | 80-1 ($\times$ 3258) | Increasing the predictor's spatial domain |
| **CNN-LM** | 20-**50**-**25**-**1**-3258 | Using convolutions to automatically obtain meaningful spatial predictors |
| **CNN1** | 20-**50**-**25**-**1**-3258 | Testing the added value of CNN non-linearity |
| **CNN10** | 20-**50**-**25**-**10**-3258 | Increasing the complexity of last CNN features layer |
| **CNN-PR** | 20-**10**-**25**-**50**-3258 | Using standard topologies from pattern recognition |
| **CNNdense** | 20-**50**-**25**-**10**-50-50-3258 | Using complex dense CNN models |

**Table 2.** Description of the deep learning architectures intercompared in this study, together with the two benchmark methods: GLM1 and GLM4 (these models are trained separately for each of the 3258 land-only gridboxes in E-OBS). Convolutional layers are indicated with boldfaced numbers. The numbers indicating the architecture correspond to the number of neurons in the different layers (in bold for convolutional layers).

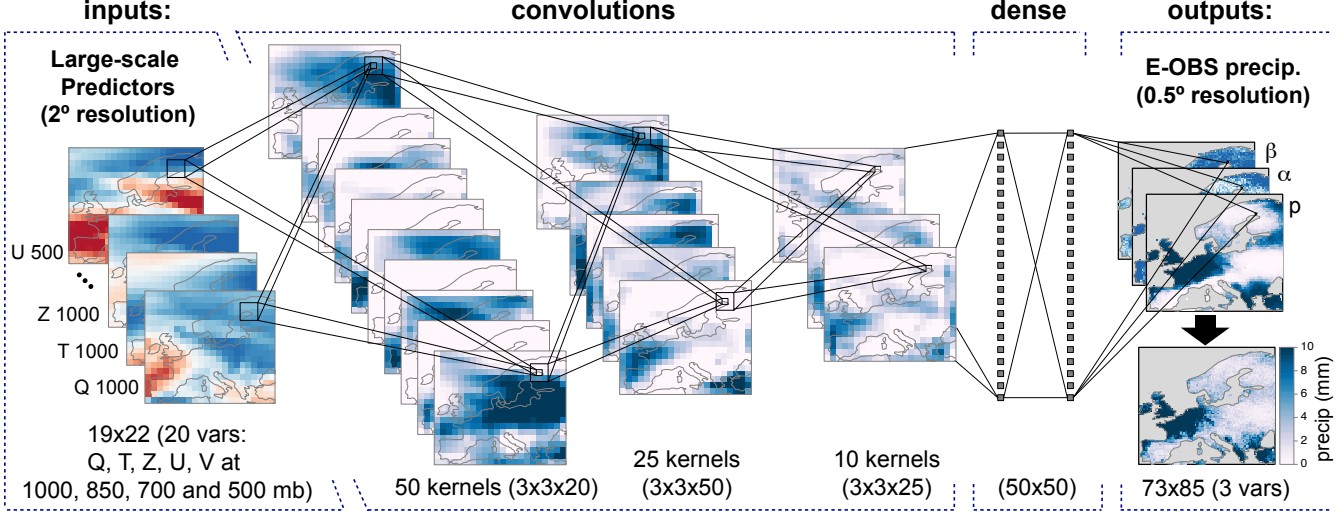

**Figure 3.** Scheme of the convoloutional neural network architecture used in this work to downscale European (E-OBS $0.5°$ grid) precipitation based on five coarse ($2°$) large-scale standard predictors (at four pressure levels). The network includes a first block of three convolutional layers with 50, 25 and 10 ($3 \times 3 \times \#inputs$) kernels, respectively, followed by two fully-connected (dense) layers with 50 neurons each. The output is modeled through a mixed binomial-lognormal distribution and the corresponding parameters are estimated by the network, obtaining precipitation as a final product, either deterministically (the expected value), or stochastically (generating a random value from the predicted distribution). The output layer is activated linearly except for the neurons associated to the parameter $p$ which present sigmoidal activation functions.

ization capability is configuration specific and thus difficult to assess in general. Recently, Reichstein et al. (2019) outlined this problem and encouraged research towards the understanding of deep neural networks in climate science. In this study we aim to

shed light on the particular role of the different elements conforming the deep neural network architecture (e.g., convolutional and fully-connected or dense layers). To do this, we build and evaluate deep SD models of increasing complexity, starting with a simple benchmark linear model (GLM) and adding additional "deep" components, in particular convolution and dense layers, as shown schematically in Figure 3.

    The basic neural network topology relies on feed-forward networks composed of several layers of non-linear neurons which
are fully-connected between consecutive layers, from the input to the output (these are commonly referred to as "dense" networks; see Figure 3). Each of these connections is characterized by a weight which is learnt from data (e.g. the two layers of 50 neurons each in Figure 3 result in a total of $50 \times 50$ internal weights, besides the input and output connections). Differently to standard dense networks (whose input is directly the raw predictor data), convolutional networks generate data-driven spatial features to feed the dense network. These layers convolute the raw gridded predictors using 3D kernels (variable, latitude and
longitude), considering a neighbourhood of the corresponding gridbox ($3 \times 3$ in this work) in the previous layer (see Figure 3). Instead of fully-connecting the subsequent layers, kernel weights are shared across regions, resulting into a drastic reduction in the degrees of freedom of the network. Due to these convolutional operations, layers consists on filter maps, which can be interpreted as the spatial representation of the feature learned by the kernel. This is crucial when working with datasets with an underlying spatial structure.

To maximize the performance of convolutional topologies, it is necessary to select an adequate number of layers, filter maps and kernel's size, which has been done here following a screening procedure testing different configurations varying mainly the number of layers (up to 6), the kernel size ($3 \times 3$, $5 \times 5$, and $7 \times 7$ kernels), and the number of neurons in the dense layer (25, 50 and 100). As a result of this screening we obtained an optimum of 3 convolutional layers and a $3 \times 3$ kernel size; moreover, the best results when including the dense final component were obtained with two layers of 50 neurons each; this
resulting configuration is displayed in Figure 3. Therefore, additional layers seem to not benefit the model due to an over-parameterization when more nonlinearity is actually not needed. Likewise, the final choice of kernel's size ($3 \times 3$) is related to the fact that this is an informative scale for downscaling at the resolution considered in this work, with larger spatial information built as a result of layer composition. Besides the different deep learning architectures, we also analyzed the effect of basic elements such as the activation function or the layer configuration, testing different configurations.

All the deep models used in this work have been trained using daily data for both predictors and predictand. For temperature, the output is the mean of a gaussian distribution (one output node for each target gridbox) and training is performed by minimizing the mean squared error. For precipitation, due to its mixed discrete-continuous nature, the network optimizes the negative log-likelihood of a Bernoulli-Gamma distribution following the approach previously introduced by Cannon (2008). In particular, the network estimates the parameter $p$ (i.e., probability of rain) of the Bernoulli distribution for rain occurrence,
and the parameters $\alpha$ (shape) and $\beta$ (scale) of the Gamma rain amount model, as illustrated in the output layer of Figure 3. The final rainfall value for a given day $i$, $r_i$, is then be inferred as the expected value of a gamma distribution, given by $r_i = \alpha_i * \beta_i$.

    The first two methods analyzed in this work are the two benchmark GLM models (i.e. multiple linear regression for temperature and Bernoulli + gamma GLM for precipitation) considering local predictors at the nearest (4 nearest) neighboring gridboxes. They are labelled as GLM1 (GLM4) in Table 2. Selecting information only from the local gridboxes could be

a limitation for the methods and, therefore, some GLM applications consider spatial features as predictors instead, such as Principal Components from the Empirical Orthogonal Functions (EOFs) (Gutiérrez et al., 2018). Convolutional networks are automatic feature extraction techniques which learn spatial features of increasing complexity from data in a hierarchical way, due to its (deep) layered-structure (LeCun and Bengio, 1995). Therefore, as third model we test the potential of convolutional layers for spatial feature extraction by considering a linear convolutional neural network with three layers (with 50, 25 and 1

features each) and linear activation functions (CNN-LM in Table 2). The benefits of non-linearity are tested considering the same convolutional network CNN-LM, but with non-linear (ReLu) activation functions in the hidden layers, making the model non-linear (CNN1 in Table 2). Moreover, the role of the number of convolutional features in the final layer is tested considering a non-linear convolutional model, but with 10 feature maps (coded as CNN10). Note that the previous models are built using a decreasing number of features in the subsequent convolutional layers. However, the approach usually used in computer vision

for pattern recognition tasks is the contrary (i.e. the number of convolutional maps increases along the network). Therefore, we also tested this type of architecture considering a convolutional neural network with an increasing number of maps, (10, 25 and 50, labelled as CNN-PR). Finally, a general deep neural network is formed by including a dense (feed-forward) network as an additional block taking input from the convolutional layer (see Figure 3). This is the typical topology considered in practical applications, which combines both feature extraction and non-linear modeling capabilities (denoted as CNNdense in Table 2).

All deep learning models listed in Table 2 have been tested with and without padding (padding maintains the original resolution of the predictors throughout the convolutional layers, avoiding the loss of information that may occur near the borders of the domain), keeping in each case the best results for the final intercomparison. Padding was found to be useful only when the amount of feature maps in the last layer was small, so padding is only used for CNN1 model.

## 4   Results

In this section we intercompare and discuss the performance of the different models shown in Table 2 for temperature (Section 4.1) and precipitation (Section 4.2).

### 4.1   Temperature

Figure 4 shows the validation results obtained for temperature in terms of the different metrics explained in Section 2.2. Each panel contains 7 boxplots, one for each of the methods considered (Table 2), representing the spread of the results along the

225 entire E-OBS grid. In particular, the gray boxes corresponds to the 25-75 percentile range, whereas the whiskers cover the 10-90 percentage range. The horizontal red line plots the median value obtained from the GLM4 method, which is considered as benchmark.

In general, all methods provide quite satisfactory results, with low biases and RMSE (panels a, d, e and f), a realistic variability (c) and very high correlation values (after removing the annual cycle from the series; panel b). Among the classic

linear methods, GLM4 clearly outperforms GLM1, which highlights the fact that including predictor information representative of a wider area around the target point helps to better describe the synoptic features determining the local temperature. However,

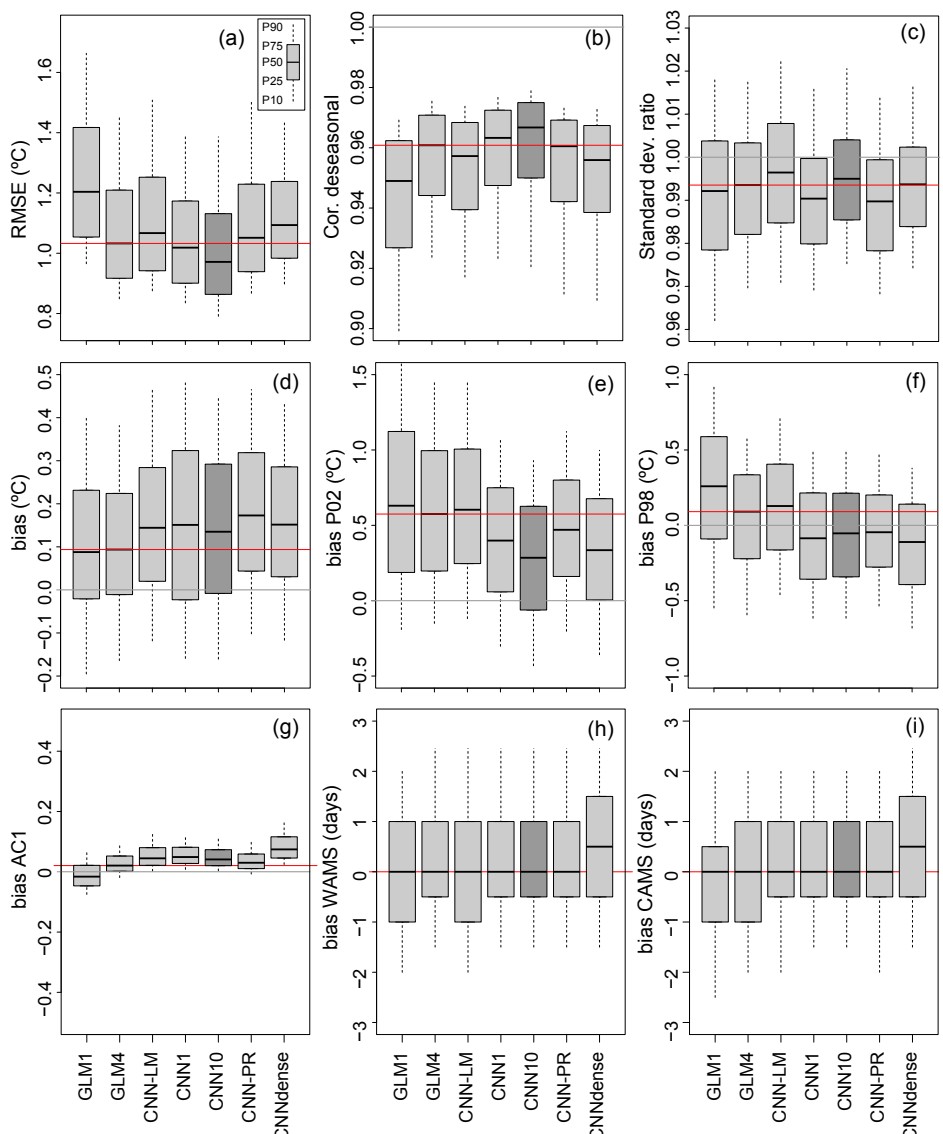

**Figure 4.** Validation results obtained for temperature. Each panel (corresponding to a particular metric) contains 7 boxplots, one for each of the methods tested, which represents the spread of the results along the entire E-OBS grid (the gray boxes corresponds to the 25-75 percentile range, whereas the whiskers cover the 10-90 percentage range). The horizontal red line plots the median value obtained from the GLM4 method, which is considered as benchmark, whereas the gray one indicates the 'perfect' value for each metric. The dark shaded box indicates the best performing method taking into account all metrics simultaneously (CNN10 in this case).

most of the local variability seems to be explained by linear predictor-predictand relationships, as both GLM4 and CNN-LM provide similar results to more sophisticated neural networks which account for non-linearity (regardless of their architecture).

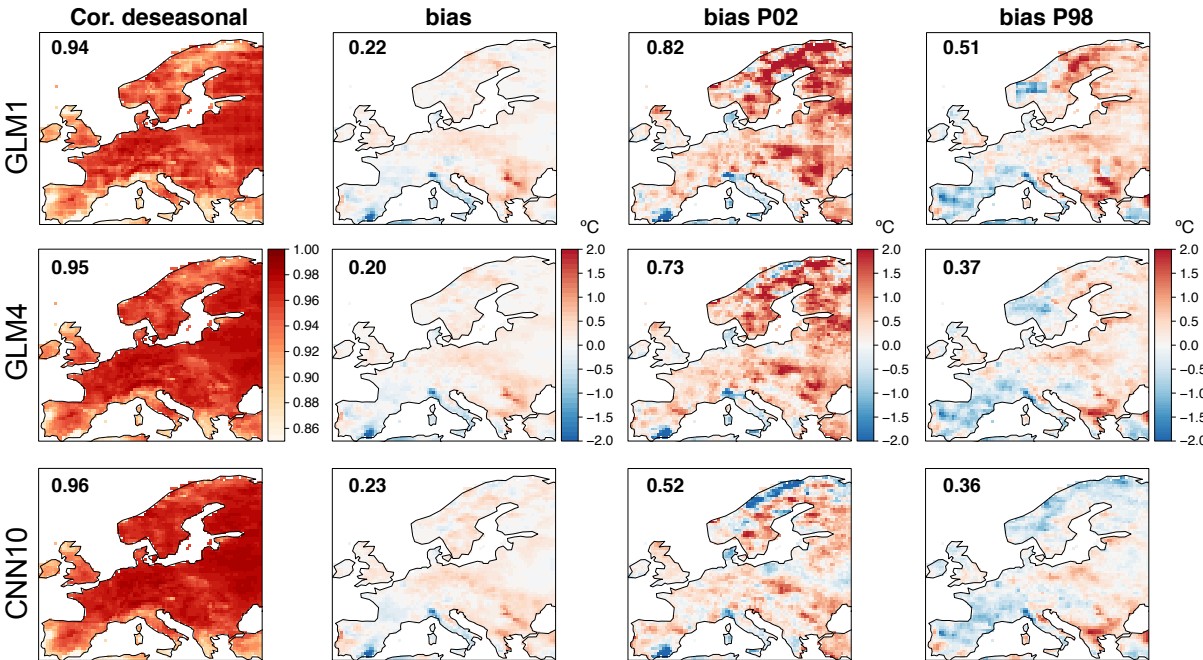

**Figure 5.** Maps showing the spatial results obtained in terms of the different metrics considered for temperature (in columns) for the two benchmarking versions of GLM (top and middle row) and the best-performing method, the CNN10 (bottom row). The numbers within the panels show the spatial mean absolute values (to avoid error compensation).

Nevertheless, the biases provided by CNN1, CNN10, CNN-PR and CNNdense for P02 and P98 are lower than those obtained
from the GLM1, GLM4 and CNN-LM (e, f), which suggests that non-linearity add some value for the prediction of extremes. Despite the addition of nonlinearity to the model, benefits of convolutional topologies also come in the ability to learn adjustable regions and overcome the restrictive limitation of considering just four neighbours as predictor data. Among the neural-based models, the CNNdense model is the worst in terms of local reproducibility. This suggest that mixing the spatial features learned with the convolutions in dense layers results into a relevant loss of spatial information affecting the downscaling.
Besides, CNN10 (identified with a darker gray) provides the lowest RMSE and the highest correlations, being overall the best method.

According to the temporal metrics computed (panels g, h and i in Fig.4) we can state that no method clearly outperforms the others in terms of reproduction of spells for temperature. Despite there is some spatial variability (spread of the boxplots) the median results are nearly unbiased in all cases (except for the CNNdense model).

For a better spatial interpretation of these results, Figure 5 shows maps for each metric (in columns) for GLM1, GLM4 and CNN10 (in rows), representing the two initial benchmarking methods and the best-performing CNN model in this case. Due to its strong local dependency, GLM1 leads to patchy (discontinuous) spatial patterns, something which is solved by GLM4 — including local predictor information representative of a wider area around the target point provides smother patterns.— Beyond this particular aspect, the improvement of GLM4 over GLM1 is evident for RMSE and correlation, and to a lesser extent also

for the bias in P98. However, the best results are found for the CNN10 method for the above mentioned particularities, which improves all the validation metrics considered, and in particular, the bias for P2. As already pointed out in Section 2.1, note that the anomalous results found for Southern Iberia could likely be related to issues in the E-OBS dataset.

It is important to highlight that the three methods present very small (mean) biases along the entire continent, which suggests their good extrapolation capability, and therefore, their potential suitability for climate change studies (recall that the
255 anomalously warm test period that has been selected for this work may serve as a surrogate of the warmer conditions that are expected due to climate change). In order to further explore this issue, we have also analyzed the capability of the models to produce extremes which are larger than those in the calibration data. To this aim we have considered the 99th percentile over the historical period as a robust reference of extreme value, and calculated the frequency of exceeding this value in the test period for the observations and the GLM1, GLM4 and CNN10 downscaled predictions. The results are shown in Figure 6
and indicate that the three models (in particular the latter two) are able to reproduce the same frequency and spatial pattern of out-of-sample days observed in the test period.

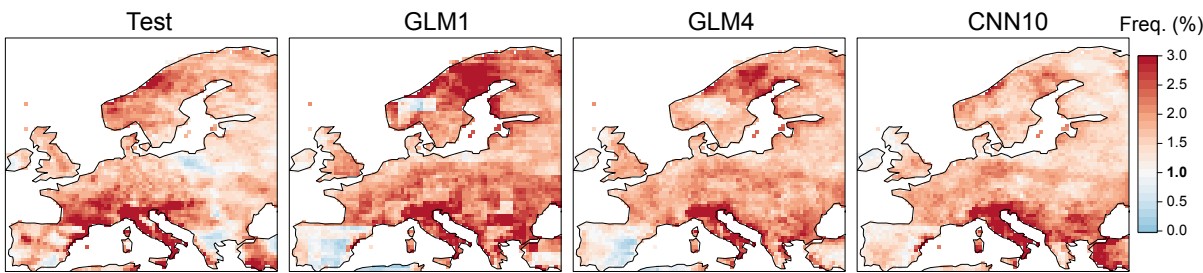

**Figure 6.** Frequency of exceeding the 99th percentile value of the training period in each of the gridboxes for the observations in the test period and the test predictions of the GLM1, GLM4 and CNN10 models (in columns). Note that a frequency of 1% (in boldface) would indicate the same amount of values exceeding the (extreme) threshold than in the training period.

## 4.2 Precipitation

Figure 7 is similar to Figure 4, but for the case of precipitation (note that the validation metrics considered for this variable differ). Similarly to the case of temperature, GLM4 performs notably better than GLM1, in particular for the ROCSS (panel
a), the RMSE (b), and the correlation (c). Note that to compute the ROCSS we use the probabilistic output of the logistic regression, for the GLM1 and GLM4 models, and the direct estimation of the parameter $p$ on the neural models. Nevertheless, with the exception of CNN-LM and CNN-PR, convolutional networks yield in general better results than GLM4. Differently to the case of temperature, the results obtained indicate that accounting for non-linear predictor-predictand relationships is key to better describe precipitation. The latter is based on the improvement of nonlinear models with respect to the linear ones
(GLM1, GLM4 and CNN-LM), especially in terms of ROCSS and correlation. Moreover, the standard architecture for pattern recognition (CNN-PR), is not suitable for this prediction problem probably due to an over-parameterization in the connection between the last hidden layer (50 feature maps) and the output layer (3 variables per gridpoint in contrast of the downscaling

of temperature where there was only 1 variable to estimate). In terms of errors (RMSE and the different biases considered), all convolutional networks perform similarly, exhibiting very small biases for the mean centered around zero. With respect to the P98, the slight underestimation shown by deterministic configurations (e) can be solved by stochastically sampling from the predicted Gamma distribution (f), but at the cost of losing part of the temporal and spatial correlation achieved by deterministic set-ups (not shown). Note that, as usual, the correlations found for all methods are much lower than those obtained for temperature, with the CNN-LM method yielding similar values to those obtained with GLM4. The existence of CNN-LM permits to marginalize the role of the convolutions on the spatial predictor data from the nonlinearity of the rest of the neural-based models. This analysis suggests that choosing the 4 nearest gridboxes as predictors allows to capture the key spatial features that affect the downscaling of precipitation with linear models (at least over Europe). Differently to the case of temperature, note also that there is not a significant change in the climatological mean between the train and test periods for precipitation (see Figure 2), so the particular train/test partition considered in this work does not allow to carry out a proper assessment of the extrapolation capability of the different methods.

Similarly to the analysis of the temperature, there is no clearly outstanding method when analyzing the spells (panels h and i of Fig.7). The GLM4 seems to be unbiased for the WetAMS, however all models tend to overestimate the DryAMS by 2-3 days on average. The GLM1 model performs clearly worse than the rest, probably due to the limited amount of predictor information involved in this method. It has to be noted in this analysis that temporal components have not been explicitly added to the models (e.g., in form of recurrent connections) neither linear nor neural ones, and therefore the reproduction of spells can be affected.

Overall, the best results are obtained for the CNN1 (marked with a darker gray) and CNNdense, which differ from CNN10 in the amount of neurons placed in the last hidden layer. This suggests that whilst one feature map was a little restrictive for the case of temperature, ten maps oversized the network for precipitation, worsening its generalization capability for this variable. The latter may be directly proportional to the number of connections in the output layer, which is dependent on the number of filter maps of the last hidden layer and on the output neurons, which is three times bigger for the downscaling of precipitation than for temperature.

Figure 8 is the equivalent to Figure 5 but for precipitation. Again, the best-performing method (CNN1 in this case; bottom row) is shown, together with the two benchmarking versions of GLM (top and middle rows). In all cases, the deterministic implementation is considered. As for temperature, GLM4 provides better results than GLM1 for all metrics, being the spatial pattern of improvement rather uniform in all cases. Likewise, CNN1 outperforms GLM4 for all metrics and regions, especially over Central and Northern Europe. These results suggest the suitability of convolutional neural networks to downscale precipitation, which may be a consequence of their ability to automatically extract the important spatial features determining the local climate, as well as to efficiently model the non-linearity established between local precipitation and the large-scale atmospheric circulation.

Finally, notice that the anomalous results found over north-eastern Iberia and the Baltic states might be due to issues in the E-OBS dataset. Nonetheless, particularly bad results are also found over the Greek peninsula (especially for the mean bias), for which we do not envisage a clear explanation.

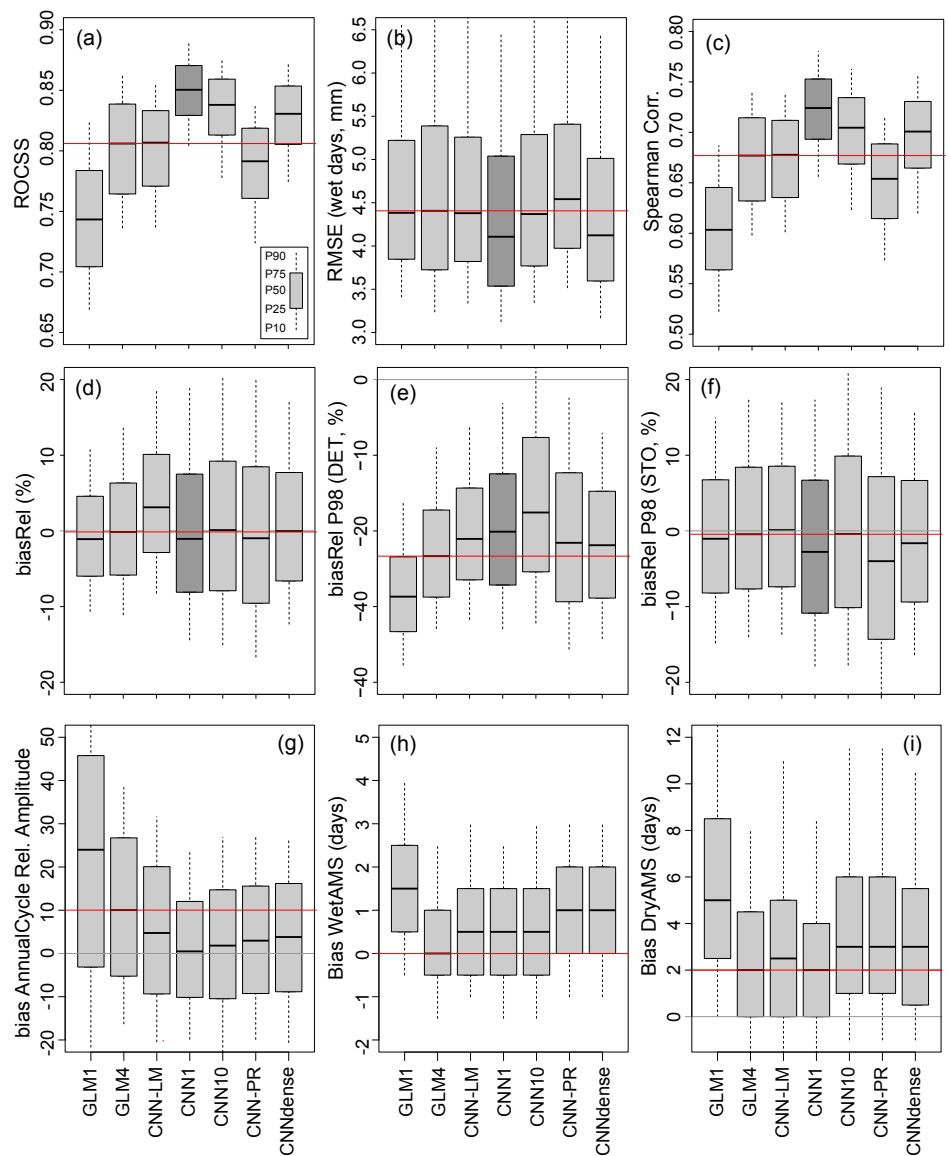

**Figure 7.** As Figure 4, but for precipitation. For the relative bias of the P98 the labels 'DET' and 'STO' refer to deterministic and stochastic, respectively.

## 5 Conclusions

Deep learning techniques have gained increasing attention due to the promising results obtained in various disciplines. In particular, convolutional neural networks (CNN) have recently emerged as a promising approach for statistical downscaling in climate due to their ability to learn spatial features from huge spatio-temporal datasets, which would allow for an efficient

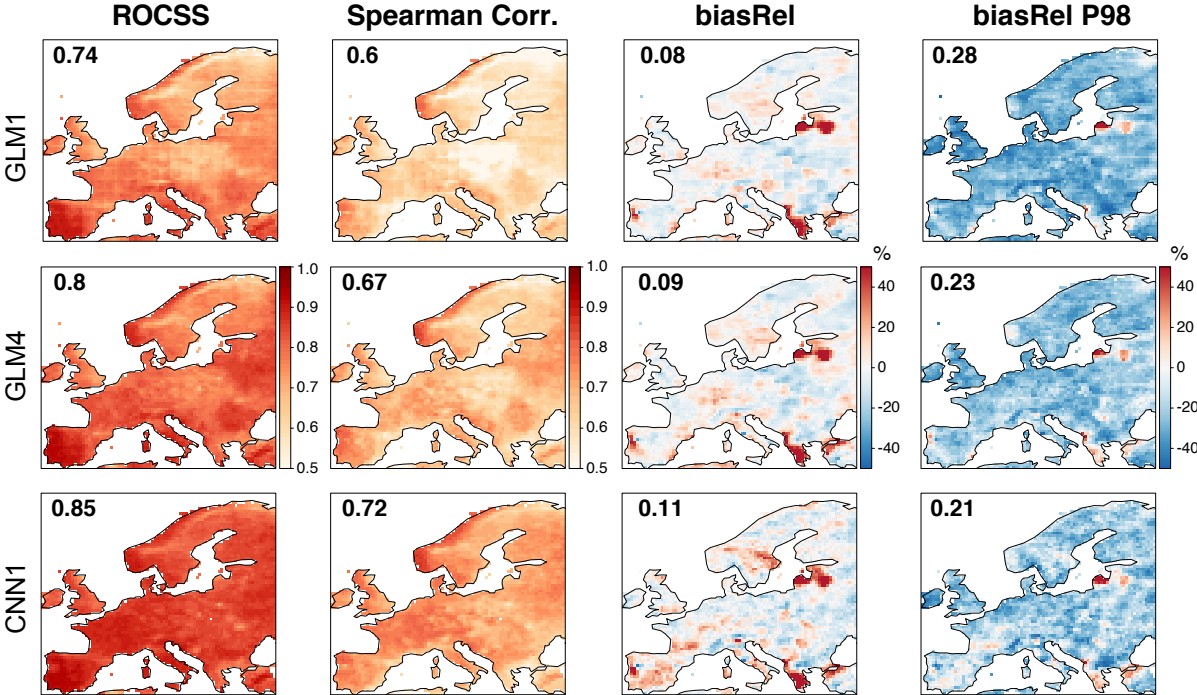

**Figure 8.** As Figure 5 but for precipitation. In this case, CNN1 is taken as best-performing method (bottom row). The numbers within the panels show the spatial mean absolute values (to avoid error compensation).

application of statistical downscaling to large domains (e.g. continents). Within this context, there have been a number of intercomparison studies analyzing standard and machine learning (including CNN) techniques. However, these studies are based on different case studies and use different validation frameworks, which makes difficult a proper assessment of the (possible) added value offered by CNNs and, in some cases, offer contradictory results (e.g. Vandal et al., 2019; Sachindra et al., 2018).

In this paper we build on a comprehensive framework for validating statistical downscaling techniques (the VALUE validation framework) and evaluate the performance of different CNN models of increasing complexity for downscaling temperature and precipitation over Europe, comparing them with a few standard benchmark methods from VALUE (linear and generalized linear models). Besides analyzing the adequacy of different network architectures, we also focus on their extrapolation capability, a critical point for their possible application in climate change studies, and use a warm test period as surrogate of possible future climate conditions.

Regarding the classic (generalized) linear methods, our results show that using predictor data in several gridboxes helps to better describe the synoptic features determining the local climate, yielding thus better predictions both for temperature and precipitation. Besides, for the case of temperature, we find that the added value of non-linear CNNs (regardless of the architecture considered) is limited to the reproduction of extremes, as most of the local variability of this variable is well captured with standard linear methods. However, convolutional topologies can handle high-dimensional domains (i.e., continental-

sized) performing an intrinsic feature reduction step in the hidden layers, avoiding tedious and somewhat limited feature selection/reduction techniques out of the learning process. The latter results in an advantage of convolutional networks over classical approaches even when the predictor-predictand link is linear. However, for temperature, mixing the spatial features learned in the dense layers (CNNdense) adds an unnecessary complexity to the newtork due to the linearity of the link, resulting into worse predictions than those obtained with the GLMs. Moreover, for precipitation, CNNs yield in general better results than standard generalized linear methods, which may reflect the ability of these techniques to automatically extract the important spatial features determining the local climate, as well as to efficiently model the non-linearity established between this variable and the large-scale atmospheric circulation. In addition, due to the dense connection to the output's layer (where for precipitation is three times bigger than for temperature), the size of the last hidden layer plays a major role in the overparameterization of the net leading to overfitted predictions when the number of filter maps is to high (e.g., CNN-PR and CNN10). For these reasons, the models CNN1 and CNN10 were found to be the 'best' topologies for the downscaling of precipitation and temperature, respectively.

It is worth to mention that any of the methods considered in this work is specifically designed to reproduce advanced temporal aspects such as spells. In the coming future, we plan to explore other battery of methods which explicitly aim to accurately reproduce the observed temporal structure, such as recurrent neural networks.

Note that the overall good results found for the CNNs tested here, together with the fact that they can be suitably applied to large domains without worrying for the spatial features being considered as predictors, can foster their use for statistical downscaling in the framework of international initiatives such as CORDEX, which has traditionally relied on dynamical simulations.

*Code availability.* For the purpose of research transparency, we provide notebooks with the full code needed to reproduce the experiments presented in this paper, which can be found in the DeepDownscaling GitHub repository https://github.com/SantanderMetGroup/ DeepDownscaling (Baño Medina et al., 2020). The code builds on the open-source `climate4R` (Iturbide et al., 2019) and `keras` (Chollet et al., 2015) R frameworks, for the benchmark and the CNN models, respectively. The former is an open R framework for climate data access, processing (e.g. collocation, binding, and subsetting), visualization, and downscaling (package `downscaleR`[ Bedia et al., 2019)), allowing for a straightforward application of wide range of downscaling methods. The latter is a popular R framework for deep learning which builds on `TensorFlow`.

Moreover, in order to facilitate the development of deep learning downscaling methods we developed an extension of the `downscaleR` package using `keras`, which is referred to as `downscaleR.keras` (https://github.com/SantanderMetGroup/downscaleR.keras) and is used for the first time in this paper (see the companion notebooks).

Moreover, the validation of the methods has been carried out with the package VALUE and its climate4R wrapper `climate4R.value` (https://github.com/SantanderMetGroup/climate4R.value), which enables a direct application of the VALUE validation metrics in the framework of `climate4R`.

## Appendix A: Computing times

In this Appendix we analyse the computation times required for the calculation of the downscaling methods used in this study. All methods build on the R framework `climate4R` (https://github.com/SantanderMetGroup/climate4R, Iturbide et al., 2019), in particular on the package `downscaleR` (Bedia et al., 2019) for the linear (GLM) benchmark models and on the package `downscaleR.keras` (presented in this study) for the new deep learning CNN models. In order to test the computational effort of the methods, we have isolated in both packages the code needed to train the models and to predict the test period. The resulting times for both generalized linear models (GLM) and deep CNN models are shown in Table A1, corresponding to the execution on a single machine with operative system Ubuntu 16.04 LTS (64 bits), with 16 GB memory and 8 processing unit Intel® Core™ i7-6700 3.40GHz.

It must be noted that for precipitation there are two GLMs to train (a binomial logistic and a gamma logarithmic for the occurrence and amount of rain, respectively) and therefore, the time included in the table for GLM1 and GLM4 is the sum of these two individual GLMs. Differently, in deep learning models the occurrence and amount of rain are trained simultaneously. In this case, the speed of training depends on some parameters such as the learning rate (learning rate equal to 0.0001 in this work) and the early-stopping criteria (patience with 30 epochs) which mainly drive the number of epochs or iterations needed to train the model; these parameters have been configured for the particular application of this paper using a screening process.

|  | GLM1 | GLM4 | CNN1 |
|---|---|---|---|
| **Precipitation** | 47 | 80 | 74 |
| **Temperature** | 22 | 28 | 58 |

**Table A1.** Computation times (in minutes) required for the calculation (training and prediction of the test period) for three downscaling methods used in this study: GLM1, GLM4, and CNN1 (the rest of deep configurations yield similar computing times).

Table A1 indicates that GLM4 is more time consuming than the simplified counterpart (GLM1) due to a larger number of predictors. Moreover, the time needed to train the deep CNN1 is similar to that required for GLM4 for precipitation (twice for temperature, in agreement with the use of a single/two models for temperature/precipitation GLMs). Therefore, the computational effort is not a strong limitation for continental-wide applications of deep learning models. The main reason for this result is that the GLMs are trained at a gridbox level (one model trained for each gridbox), whereas the CNN is naturally multisite; therefore, although the training is very time consuming, a single CNN model is needed for the whole domain. However, note that for smaller domains (e.g. national-wide) the difference between GLMs and CNNs could be large (the computation time of GLMs decreases linearly with the number of gridboxes) and could make a difference.

*Author contributions.* Baño-Medina, J. and Gutiérrez, J.M. conceived the study; Baño-Medina, J. implemented the code to develop the convolutional neural networks, and generated the results of the paper; all authors analyzed the results and wrote the manuscript; Baño-Medina, J. and Manzanas, R. prepared the code and notebooks for reproducibility.

*Competing interests.* No competing interests are present.

*Acknowledgements.* The authors acknowledge the funding provided by the project MULTI-SDM (CGL2015-66583-R, MINECO/FEDER). We also acknowledge the E-OBS dataset from the EU-FP6 project UERRA (http://www.uerra.eu) and the Copernicus Climate Change Service, and the data providers in the ECA&D project (https://www.ecad.eu).

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
