# Peer review of "Configuration and Intercomparison of Deep Learning Neural Models for Statistical Downscaling"

_Geoscientific Model Development, 2019_

## Referee Comment (RC1) · Matteo De Felice (Referee) · 21 Nov 2019

In literature we can find several works about downscaling using statistical methods or machine learning methods: on this topic there are many questions still to answer. This paper contributes with two main positive points: 1. it's part of a rigorous project/experiment framework (VALUE) amd 2. it is fully reproducible (from the data to the algorithms, available on Github and Zenodo). I have a few comments that I think would improve the quality of the paper:

1. The authors should say something about the computational effort of the proposed methods, saying if there is any trade-off between performances (RMSE, correlation,

etc.) and complexity/computation time. I expect that a linear model should run much faster than a CNN, can the authors say something about this? 2. Possibly related to the point 1. probably: the authors use different CNN setups but then they analyse only the best one (CNN1), can they say something about the others? Why they do not work well? Why they were supposed to work well? Why they are considered in the paper? 3. For the precipitation the authors use a probabilistic score (ROCSS) in addition to the common ones (RMSE, Correlation, etc), it's not clear how the output of a linear model or a CNN could be considered a probabilistic forecast. They should clarify this point. 4. Can the authors comment (or provide reference) on how they decided the best configuration for the CNN? Number of layers, etc. This could be beneficial especially considered that the journal is for a community that, as you say, does not really trust deep learning models. 5. Regarding the comment about deep learning and distrust in climate community, I have the impression that the problem is not just about the extrapolation capabilities, but in general about the impossibility to really know how a black-box model operates. The extrapolation is only a part of it. You can not really assess the capability to "extrapolate" for a complex model like a CNN because any assessment would be 1. configuration specific and 2. data specific. Then I think the problem is a conceptual one: the difficulty in generalising the behaviour of a very complex and highly-nonlinear model. (This point is just a personal comment, I think that this paper is not the right place to for this kind of discussion however I have really appreciated that comment) 6. Can the authors provide a map with the difference between metrics (RMSE or correlation) between GLM4 and CNN1? Can they say something about the areas where CNN/GLM outperforms the other method? 7. The DOI at line 72 does not work 8. There is a typo in the first panel of Figure 1, in the caption title.

---

## Referee Comment (RC2) · Anonymous Referee #2 · 17 Dec 2019

This study investigates the ability of convolutional neural networks (CNN) for downscaling daily temperature and precipitation over Europe. The added value is found for extreme temperature and most metrics for precipitation. The paper is well written and the results demonstrate the importance of accounting for non-linearity for downscaling precipitation. I have a few concerns that the authors should be able to address.

Major comments 1. It would be good to say something more specifically about European applications and the related data needs. Climate change studies are mentioned in the text. But nothing is said about the types of applications/users in the Introduction – and this influences the types of information required – e.g., whether spatial consistency

is important, the types of extremes that are relevant.

2. Line 33-39: This study focuses on the deep learning techniques in the context of perfect prognosis SD. However, it's not clear what the difference between classical SD methods and machine learning techniques is. This needs to be mentioned in the introduction.

3. A warm validation period is selected as surrogate of possible future climate conditions to investigate the suitability of CNN in climate change studies. It should be better to clearly state how the models produce extremes which are larger than those in the calibration data, and the ability of models to account for changes in the statistics in the future (related to the stationarity assumption).

4. Different network architectures of CNN have been evaluated and intercompared in this study. However, the authors should provide more interpretations on the impact of these configuration on model performance. There are a few examples where this is currently done (e.g., lines 215-218, 238-244) but this needs to be done more systematically, and highlighted in the conclusion section.

5. The skill of the various downscaling methods is assessed mostly on spatial variability. How could the CNN reproduce the temporal variability of the local climate? You may want to validate the ability of CNN to represent dry/wet spells and interannual variation.

Minor comments 1. Line 13: What does 'classic ones' refer to? Need to make them clear. 2. Line 79: 'such'  $\rightarrow$  'such as' 3. Line 111: 'vale' should be 'value'. 4. Figure 2: The label 'bias' is misleading here, since the map shows the differences between the test and train periods based on observations. 5. Figure 4 & 6: The best method is in fact different for each metric, but the same best method (CNN10 for temperature and CNN1 for precipitation) for all metrics is indicated in the figure. How do you choose the best performing method, may be based on one metric? 6. Figure 6: Please explain 'DET'(e) and 'STO'(f). 7. Traditional statistical downscaling methods generally require high-resolution obserbations for model training, thus it is difficult to provide downscaled

cliamte simulations for the regions with little observation data. Is the skill of CNN sensitive to the resolution of observations?

СЗ

---

## Author Comment (AC1) · 28 Dec 2019

**1**. The authors should say something about the computational effort of the proposed methods, saying if there is any trade-off between performances (RMSE, correlation, etc.) and complexity/computation time. I expect that a linear model should run much faster than a CNN, can the authors say something about this?

We have the of packages used set climate4R (https://github.com/SantanderMetGroup/climate4R) for the linear models downscaleR.keras and the package (https://github.com/SantanderMetGroup/downscaleR.keras), which integrates keras into climate4R, for the deep models. We have isolated the code needed to train and predict in the test set for both generalized linear models (GLM) and deep models and retained the computation times (see the table below). It must be noted that for precipitation there are two generalized linear models to train (a binomial logistic and a gamma logarithmic for the occurrence and amount of rain, respectively) and therefore, the time included in the table for GLM1 and GLM4 is the sum of these two individual GLMs. Differently, in deep learning models the occurrence and amount of rain are trained simultaneously. In this case, the speed of training is very sensitive to some parameters such as the learning rate (learning rate equal to 0.0001) and the early-stopping criteria (patience with 30 epochs) which mainly drive the number of epochs or iterations needed to train the model. Taking into account all these considerations we observe little difference between the computational times needed to train the linear models (GLM1 and GLM4) and the deep ones (CNN1; the rest of deep configurations yield similar computing times). Therefore, the computational effort required to train and run deep models is not a strong limitation.

|               | GLM1 | GLM4 | CNN1 |
|---------------|------|------|------|
| precipitation | 47   | 80   | 74   |
| temperature   | 22   | 28   | 62   |

Computational times (in minutes) needed to train the GLM1, GLM4 and CNN1 methods (see Table 2 of the manuscript for information about the configuration of the models) for precipitation and temperature.

Based on this referee's comment we have included an annex in the new version of the manuscript which shows the computational times required by the different methods.

**2. Possibly related to the point 1. probably: the authors use different CNN setups but then they analyse only the best one (CNN1), can they say something about the others? Why they do not work well? Why they were supposed to work well? Why they are considered in the paper?**

Unlike other disciplines where deep learning is well established, the "black box" character of neural networks is a major concern in the earth sciences community. In contrast to other deep learning and downscaling studies where complex computer vision topologies are adopted without a proper justification, in this study we propose an intercomparison among deep models of increasing levels of complexity in order to shed light on the role of the different elements involved in this kind of approaches for downscaling. For instance, we consider a convolutional model with linear activation functions (CNN-LM in the manuscript) and its equivalent with nonlinear activation functions (CNN1). This allows to analyze the influence of nonlinearity in the downscaling. Our results show that the introduction of nonlinearities in the model is relevant for precipitation but not for temperature. CNNdense and CNN-PR (see Table 2 of the manuscript for the details) were included in the study since this type of networks are often used in computer vision applications and we wanted to test their potential suitability for statistical downscaling purposes. Whereas CNNdense results from the idea of mixing the spatial patterns learn by the convolutions in the last hidden layers, CNN-RPR is based on the idea that more filter maps are needed as we go further in the net, given the increase of nonlinearity. Both methods obtain similar results to GLM4 and are clearly outperformed by only-convolutional topologies due to the spatial dependence of the predictand's output neurons in the last hidden layer (i.e., the prediction over a particular site its dependent on the atmospheric situation surrounding that area and thus mixing the spatial patterns in dense layers damages the downscaling).

A description of the deep models proposed and why they are considered in the current paper (lines 157-170) can be found in Section 3 of the manuscript. All deep learning models have been intercompared in Figures 4 and 6 in terms of all the proposed metrics for temperature (RMSE, Pearson correlation, bias of the mean, percentile 2 and percentile 98 and the ratio of standard deviations) and for precipitation (ROCSS, RMSE, Spearman correlation, bias for the mean and percentile 98). The spatial maps displayed in Figures 5 and 7 were only shown for the generalized linear models (GLM1 and GLM4) and for the best method (CNN1 and CNN10 for precipitation and temperature, respectively) as no additional conclusions than those inferred from Figures 4 and 6 appeared when considering the other deep models.

**3**. For the precipitation the authors use a probabilistic score (ROCSS) in addition to the common ones (RMSE, Correlation, etc), it's not clear how the output of a linear model or a CNN could be considered a probabilistic forecast. They should clarify this point.**

Note that the ROCSS in only used for the binary (0/1) event *occurrence of precipitation*. In GLMs, there is a first GLM with binomial error distribution and logit link function whose outputs can be directly understood as probability of rain for a given day at a given gridbox.

Likewise, CNNs minimize the negative-log-likelihood of a Bernouilli-Gamma distribution (see Figure 3 and lines 170-178) providing therefore an estimation of the parameters *p*, *alpha* (shape parameter of a gamma distribution) and *beta* (scale parameter of a gamma distribution), simultaneously. *p* is the probability of rain for a given day at given gridbox. Therefore, both GLM and CNN models provide the needed information to compute the ROCSS.

More information can be found in the paper notebook (2019\_deepDownscaling\_GMD.pdf), where there is a step-to-step explanation of the results presented in the paper.

4. Can the authors comment (or provide reference) on how they decided the best configuration for the CNN? Number of layers, etc. This could be beneficial especially considered that the journal is for a community that, as you say, does not really trust deep learning models.

The number of layers depends mainly on two aspects: the degree of nonlinearity you want to achieve in your model (the deeper the more nonlinear) and the number of parameters involved in your model. Unlike computer vision applications in which there are usually more than 50.000 images available for the training phase, here we only had 24 years of daily data. As a consequence, the depth of our networks is limited. Though it was not discussed in the manuscript we tried different topologies that varied mainly in the number of layers (up to 6) and in the kernel size (we used for the paper 3x3 kernels but also tried 5x5 and 7x7 sizes). After this trial and error procedure we ended up with the optimum of 3 convolutional layers and a 3x3 kernel size. Therefore, additional layers seem to not benefit the model due to an overparameterization when no more nonlinearity is actually needed. Likewise, the final choice of kernel's size being equal to 3 is related to the fact that most relevant phenomena for downscaling at the resolution considered in this work occurs in a surrounding domain of 3x3, with bigger domains just adding unnecessary degrees of freedom to the model.

5. Regarding the comment about deep learning and distrust in climate community, I have the impression that the problem is not just about the extrapolation capabilities, but in general about the impossibility to really know how a black-box model operates. The extrapolation is only a part of it. You can not really assess the capability to "extrapolate" for a complex model like a CNN because any assessment would be 1. configuration specific and 2. data specific. Then I think the problem is a conceptual one: the difficulty in generalising the behaviour of a very complex and highly-nonlinear model. (This point is just a personal comment, I think that this paper is not the right place to for this kind of discussion however I have really appreciated that comment)

Thanks for your comment, we find interesting your discussion about the "black box" nature of neural networks. It is true that the extrapolation capability is only a part of it and further efforts have to be done with regards to other issues such as as the quantification of uncertainty in the predictions by (deep)bayesian approaches or the sensitivity to the choice of predictors.

**6. Can the authors provide a map with the difference between metrics (RMSE or correlation) between GLM4 and CNN1? Can they say something about the areas where CNN/GLM outperforms the other method?**

The figure below shows the differences in de-seasonalized correlation found between CNN1 and GLM4 for precipitation (left column) and temperature (right column). Red (blue) colors indicate that CNN1 yields higher (lower) correlations than GLM4.

For precipitation, better results are found for CNN1 over most of Europe, especially in Escandinavia, the British isles and central Europe. Eastern Europe and the Mediterranean differ slightly among models. This may be due to the fact that the "true" predictor-predictand link is quasi-linear in those areas and CNN has little added value apart from the automatic treatment of the input space. This conclusion is also applicable to the results found for temperature, for which the "true" linear existing relationship is again (quasi-linear and CNN1 and GLM4 do not show significant differences in terms of correlation.

Differences in de-seasonalized correlation for precipitation (left) and temperature (right) found for CNN1 and GLM4 (the latter is taken as reference).

**7. The DOI at line 72 does not work. There is a typo in the first panel of Figure 1, in the caption title.**

We have updated the version and the correct DOI will appear in the new version of the manuscript.

---

## Author Comment (AC2) · 28 Dec 2019

**MAJOR COMMENTS:**

**1. It would be good to say something more specifically about European applications and the related data needs. Climate change studies are mentioned in the text. But nothing is said about the types of applications/users in the Introduction –and this influences the types of information required – e.g., whether spatial consistency is important, the types of extremes that are relevant.**

Sectoral studies, such as hydrology, agriculture or energy applications, are in need of high-resolution climate/meteorological information at different time scales. For example at short scales (i.e, hourly and daily), accurate forecasts of wind fields are crucial to predict the capacity of renewable sources to meet the demands of the energy market. At longer scales, the impact and adaptation communities derive indices from the downscaled climate projections to evaluate the influence of climate change in different environments (e.g., health, agriculture). In addition, certain applications are sensitive to the spatial consistency of the downscaled information (e.g., to evaluate the impacts on water resources over a certain area) or to their suitability to accurately reproduce extremes (e.g., droughts and floods can cause devastating damages in agriculture).

We have included the above paragraph in the new version of the manuscript.

**2. Line 33-39: This study focuses on the deep learning techniques in the context of perfect prognosis SD. However, it's not clear what the difference between classical SD methods and machine learning techniques is. This needs to be mentioned in the introduction.**

By classical statistical downscaling techniques we refer to traditional and well established approaches adopted by the climate community, including generalized linear models, analogs and model output statistics, but also bias correction. In the machine learning paradigm there are more sophisticated approaches such as random forests, neural networks and

support vector machines (among others). We have introduced a clarification concerning this matter in the new version of the manuscript.

**3. A warm validation period is selected as surrogate of possible future climate conditions to investigate the suitability of CNN in climate change studies. It should be better to clearly state how the models produce extremes which are larger than those in the calibration data, and the ability of models to account for changes in the statistics in the future (related to the stationarity assumption)**

The choice of a warm test period was done in order to perform a preliminary analysis of the extrapolation capabilities of the statistical models. Figures 4 and 6 in the manuscript suggest that the models intercompared are able to work under unseen conditions during the training phase since the local variability was well reproduced according to the metrics evaluated, especially with the convolutional models (with overall unbiased statistics). We are currently testing the suitability of deep learning approaches to downscale future climate scenarios provided by Global Circulation Models (GCM) and will perform a more detailed analysis related to the stationarity assumption in a future paper. Note that such an analysis is out of the scope of the present paper.

However, to address the referee's concern about the reproducibility of extremes larger than those observed in the training data, we have computed the root mean squared error (RMSE) for those days in the test set for which the observed values were higher than the percentile 95th in the train set, per gridbox. The left (right) column in the figure below shows the RMSE differences between the CNN1 and GLM4 for the case of precipitation (temperature). Red (blue) colors indicate lower RMSE values for the GLM4 (CNN1). For precipitation, CNN1 yields in general lower RMSE values (particularly over southern UK), finding only better results for GLM4 over a limited region in southeast Europe. For temperature, the situation is in general neutral, finding only regional differences in Escandinavia, where CNN1 yields lower RMSE than GLM4. The differences in the RMSE obtained between the methods can be explained

according to whether or not the predictor-predictand link benefits from nonlinearity in the reproduction of extremes.

[Figure]

*Differences in the root mean squared error (RMSE) between the CNN1 and GLM4 models (see Table 2 of the manuscript for details in their model setup) for the days in the test set (2003-2008) that have observed values higher than the percentile 95th in the train set (1979-2002) per gridbox. The left (right) column corresponds to precipitation (temperature).*

**4. Different network architectures of CNN have been evaluated and intercompared in this study. However, the authors should provide more interpretations on the impact of these configuration on model performance. There are a few examples where this is currently done (e.g., lines 215-218, 238-244) but this needs to be done more systematically, and highlighted in the conclusion section.**

The differences found in model performance among the deep learning models intercompared in this study depend on the activation function (i.e., linear or nonlinear) and/or the nature of the last hidden layer (i.e., convolutional or dense). For temperature, the predictor-predictand relationship is (quasi)linear and therefore the activation function do not influences the downscaling. In this case, the differences among models are related to the last layer's type of connection: convolutional or dense. For precipitation, the presence of non-linear activation functions benefits the downscaling.

We have highlighted this throughout the new version of the manuscript, with special care in the conclusions section.

**5.** **The skill of the various downscaling methods is assessed mostly on spatial variability. How could the CNN reproduce the temporal variability of the local climate? You may want to validate the ability of CNN to represent dry/wet spells and interannual variation.**

Apart from the correlation already shown in Figures 4 and 6 of the original manuscript, we have included the figure below in the new version of the manuscript in order to address this comment. This figure shows three temporal validation metrics for each target variable: temperature and precipitation. For temperature we show the autocorrelation lag-1 and the bias in the length of the longest warm and cold annual spells (first row, from left to right). For precipitation we show the relative amplitude of the annual cycle and the relative bias in the length of the longest dry and wet annual spells (second row, from left to right).

According to these results we can state that no method clearly outperforms the other in terms of reproduction of spells, both for temperature and precipitation. Despite there is some spatial variability (spread of the boxplots) the median results are nearly unbiased in all cases. Only the GLM1 model performs slightly worse for precipitation, which is probably due to the limited amount of predictor information involved in this method. This would indicate that spatial information in the input space is crucial to better reproduce the local variability, which has been already mentioned around Figures 4 and 6 of the original manuscript.

It is worth to mention that any of the methods considered in this work is specifically designed to reproduce advanced temporal aspects such as spells. In the coming future, we plan to explore other battery of methods which explicitly aim to accurately reproduce the observed temporal structure.

[Figure]

*Temporal validation metrics computed for temperature and precipitation (top and bottom row, respectively). For temperature, the autocorrelation lag-1 (AC1), and the bias for the length of the longest (bias WAMS) and cold (bias CAMS) annual spells are shown. For precipitation we show the relative amplitude of the annual cycle and the relative bias for the length of the longest wet (biasRel wetAMS) and dry (biasRel DryAMS) annual spells.*

**MINOR COMMENTS:**

**1. Line 13: What does 'classic ones' refer to? Need to make them clear.**

**2. Line 79: 'such'→'such as'**

**3. Line 111: 'vale' should be 'value'.**

**4. Figure 2: The label 'bias' is misleading here, since the map shows the differences between the test and train periods based on observations.**

We have addressed the minor comments 1, 2, 3 and 4 indicated by the reviewer in the revised manuscript.

**5. Figure 4 & 6: The best method is in fact different for each metric, but the same best method (CNN10 for temperature and CNN1 for precipitation) for all metrics is indicated in the figure. How do you choose the best performing method, may be based on one metric?**

Figures 4 and 6 show the validation results obtained for temperature and precipitation, respectively. For temperature, the CNN10 is the best method according to the RMSE and the de-seasonalized Pearson correlation while keeping unbiased predictions for the mean, and percentiles 2th and 98th. A similar situation occurs for the CNN1 model for precipitation, for which this method outperforms the others in terms of ROCSS, RMSE and Spearman correlation while getting good results for the rest of metrics. For these reasons we chose the CNN10 and CNN1 models to be the 'best' for temperature and precipitation, respectively.

**6. Figure 6: Please explain 'DET'(e) and 'STO'(f).**

'DET' refers to deterministic and 'STO' to stochastic. We have clarified it in the new version of the manuscript.

**7. Traditional statistical downscaling methods generally require high-resolution observations for model training, thus it is difficult to provide downscaled climate simulations for the regions with little observation data. Is the skill of CNN sensitive to the resolution of observations?**

To date we have only used deep learning to downscale to resolutions of 0.5º and, despite the sensitivity of downscaling to the observational reference considered is a relevant topic of study, it is out of the scope of this paper. However, we hypothesize that the sensitivity of the downscaling to the predictand's resolution is mainly related to the explicability of the local scale by the predictor's domain rather than by the downscaling method itself (e.g., convective precipitation is not explicable by large-scale predictors and therefore the ability to establish a robust link between both is independent of the statistical method of choice). The benefits of convolutional approaches, such as the ability to treat high-dimensional domains without previous feature selection techniques and the ability to

extract non-linear patterns from data, are intrinsic to the multisite-convolutional nature and therefore the latter skills are expected to be preserved indistinctly of the predictand's resolution. In fact, downscaling to higher resolutions may require a higher degree of nonlinearity and therefore, the skill of deep learning could be even increased in comparison with classical approaches.

In the case of regions with scarce observations, computer vision applications have benefited from a concept called "transfer learning". The idea behind transfer learning is that hidden features learned in a particular task *A* are useful in a similar task *B* and therefore, the trained network *A* (or the first hidden layers) can be used to predict task *B*. In the case of downscaling, though this has not yet been tested to our knowledge, a net trained over a well observed region (e.g., Europe) could be transferable as a pretrained-net over areas with less observations available (e.g., Arctic). Though there are still questions to be answered in this topic such as whether the hidden features learned to downscale temperature over Europe (even the most simple ones located in the first hidden layers) would be helpful to downscale in regions with scarce observations which can present their own climatic particularities.

---

## Author Response (AR1)

**Responses to the Reviewers:**

**Configuration and Intercomparison of Deep Learning Neural Models for Statistical Downscaling by J. Baño-Medina, R. Manzanas, and J.M. Gutiérrez. GMD Discussion, https://doi.org/10.5194/gmd-2019-278**

**Reviewer 1**

1. The authors should say something about the computational effort of the proposed methods, saying if there is any trade-off between performances (RMSE, correlation, etc.) and complexity/computation time. I expect that a linear model should run much faster than a CNN, can the authors say something about this?

**Response:** In the original manuscript we used the R framework *climate4R*[1] *(in particular the package downscaleR)* for the linear benchmark models and Keras for the the new CNN models. In this revised version, Keras has been integrated with the downscaleR package in order to use the data preparation and preprocessing functionalities of the package, thus simplifying the code (e.g. the companion notebooks) and facilitating the development of CNN models for downscaling. The resulting package *downscaleR.keras*[2] (presented in the revised paper), allows for a more direct comparison of the standard benchmark models included in downscaleR and the new CNN models included in downscaleR.keras. In order to test the computational effort of the methods, we have isolated in both packages the code needed to train the models and to predict the test period. The resulting times for both generalized linear models (GLM) and deep CNN models are shown in the Table below.

**Table A1.** Computation times (in minutes) required for the calculation (training and prediction of the test period) for three downscaling methods used in this study: GLM1, GLM4, and CNN1 (the rest of deep configurations yield similar computing times).

|  | GLM1 | GLM4 | CNN1 |
|---|---|---|---|
| **Precipitation** | 47 | 80 | 74 |
| **Temperature** | 22 | 28 | 58 |

It must be noted that for precipitation there are two GLMs to train (a binomial logistic and a gamma logarithmic for the occurrence and amount of rain, respectively) and therefore, the time included in the table for GLM1 and GLM4 is the sum of these two individual GLMs. Differently, in deep learning models the occurrence and amount of rain are trained simultaneously. In this case, the speed of training depends on some parameters such as the learning rate (learning rate equal to 0.0001 in this work) and the early-stopping criteria (patience with 30 epochs) which mainly drive the number of epochs or iterations needed to train the model; these parameters have been configured for the particular application of this paper using a screening process.

Taking into account all these considerations, we observe little difference between the computational times needed to train the linear models (GLM1 and GLM4) and the deep ones (CNN1); the rest of deep configurations yield similar computing times). Therefore,
* * *
[1] https://github.com/SantanderMetGroup/climate4R
[2] https://github.com/SantanderMetGroup/downscaleR.keras

the computational effort required to train and run deep models is not a strong limitation for continental-wide applications. The main reason for this result is that the GLMs are trained at a gridbox level (as many models as gridboxes), whereas the CNN is naturally multisite and, therefore, although the training is very time consuming, a single model is needed for the whole domain. Note, therefore, that for smaller domains (e.g. national-wide) the difference between GLMs and CNNs could be higher and could make a difference.

Based on this comment of the reviewer we have included this information as an annex in the revised version of the manuscript which shows the computational times required by the different methods.

2. Possibly related to the point 1. probably: the authors use different CNN setups but then they analyse only the best one (CNN1), can they say something about the others? Why they do not work well? Why they were supposed to work well? Why they are considered in the paper?

**Response:** Unlike other disciplines where deep learning is well established, the "black box" character of neural networks is a major concern in the earth sciences community. In contrast to other deep learning and downscaling studies where complex computer vision topologies are adopted without a proper justification, in this study we propose an intercomparison among deep models of increasing levels of complexity (starting with the simplest one) in order to shed light on the role of the different elements involved in this kind of approaches for downscaling. For instance, we consider a convolutional model with linear activation functions (CNN-LM in the manuscript) and its equivalent with nonlinear activation functions (CNN1). This allows to analyze the influence of nonlinearity in the downscaling. Our results show that the introduction of nonlinearities in the model is relevant for precipitation but not for temperature. CNNdense and CNN-PR (see Table 2 of the manuscript for the details) were included in the study since this type of networks are often used in computer vision applications and we wanted to test their potential suitability for statistical downscaling purposes. Whereas CNNdense results from the idea of mixing the spatial patterns learn by the convolutions in the last hidden layers, CNN-RPR is based on the idea that more filter maps are needed as we go further in the net, given the increase of nonlinearity. Both methods obtain similar results to GLM4 and are clearly outperformed by only-convolutional topologies due to the spatial dependence of the predictand's output neurons in the last hidden layer. All these models have been configured applying a screening to the involved parameters and using the values yielding optimum results.

A description of the deep models proposed and why they are considered in the current paper can be found in Section 3 of the manuscript. All deep learning models have been intercompared in Figures 4 and 6 in terms of all the proposed metrics for temperature (RMSE, Pearson correlation, bias of the mean, percentile 2 and percentile 98 and the ratio of standard deviations) and for precipitation (ROCSS, RMSE, Spearman correlation, bias for the mean and percentile 98). The spatial maps displayed in Figures 5 and 7 were only shown for the generalized linear models (GLM1 and GLM4) and for the best method (CNN1 and CNN10 for precipitation and temperature, respectively) as no additional conclusions than those inferred from Figures 4 and 6 appeared when considering the other deep models.

**3.** For the precipitation the authors use a probabilistic score (ROCSS) in addition to the common ones (RMSE, Correlation, etc), it's not clear how the output of a linear model or a CNN could be considered a probabilistic forecast. They should clarify this point.

**Response:** Note that the ROCSS in only used for the binary (0/1) event *occurrence of precipitation*. In GLMs, there is a first GLM with binomial error distribution and logit link function (with values bounded between zero and one) whose outputs can be directly interpreted as probability of rain for a given day at a given gridbox.

Likewise, CNNs minimize the negative-log-likelihood of a Bernoulli-Gamma distribution (see Figure 3 second and third paragraphs) providing therefore an estimation of the parameters *p*, *alpha* (shape parameter of a gamma distribution) and *beta* (scale parameter of a gamma distribution), simultaneously. *p* is the probability of rain for a given day at given gridbox. Therefore, both GLM and CNN models provide the information needed to compute the ROCSS.

More information can be found in the paper companion notebook (2019_deepDownscaling_GMD.pdf), where there is a step-to-step explanation of the results presented in the paper.

**4.** Can the authors comment (or provide reference) on how they decided the best configuration for the CNN? Number of layers, etc. This could be beneficial especially considered that the journal is for a community that, as you say, does not really trust deep learning models.

**Response:** As already mentioned in the paper we tested several standard configurations including convolutional and/or dense layers. The number of layers (particularly for the dense layers) defines the degree of nonlinearity achieved by the model (the deeper the more nonlinear) and also the number of parameters involved. Though it was not discussed in the original manuscript for each topology we tried different configurations varying mainly the number of layers (up to 6), the kernel size (we used for the paper 3x3 kernels but also tried 5x5 and 7x7 sizes), and the number of neurons in the dense layer. After this trial and error screening procedure we ended up with the optimum of 3 convolutional layers and a 3x3 kernel size; moreover, the best results when including the dense final component were obtained with two layers 50x50. Therefore, additional layers seem to not benefit the model due to an over-parameterization when no more nonlinearity is actually needed. Likewise, the final choice of kernel's size being equal to 3 is related to the fact that this is the minimum scale for most relevant phenomena for downscaling at the resolution considered in this work, with bigger domains built up as a result of layer composition.

This information has been included in the revised manuscript (section 3, third paragraph).

**5.** Regarding the comment about deep learning and distrust in climate community, I have the impression that the problem is not just about the extrapolation capabilities, but in general about the impossibility to really know how a black-box model operates. The extrapolation is only a part of it. You can not really assess the capability to "extrapolate" for a complex model like a CNN because any assessment would be 1.

configuration specific and 2. data specific. Then I think the problem is a conceptual one: the difficulty in generalising the behaviour of a very complex and highly-nonlinear model. (This point is just a personal comment, I think that this paper is not the right place to for this kind of discussion however I have really appreciated that comment)

**Response:** Thank you for the comment, we find interesting your discussion about the "black box" nature of neural networks. We agree that the extrapolation capability is only a part of it and we have briefly extended this discussion along the lines the reviewer mention.

6. Can the authors provide a map with the difference between metrics (RMSE or correlation) between GLM4 and CNN1? Can they say something about the areas where CNN/GLM outperforms the other method?

**Response:** The figure below shows the differences in de-seasonalized correlation found between CNN1 and GLM4 for precipitation (left column) and temperature (right column). Red (blue) colors indicate that CNN1 yields higher (lower) correlations than GLM4.

For precipitation, better results are found for CNN1 over most of Europe, especially in Escandinavia, the British isles and central Europe. Eastern Europe and the Mediterranean exhibit more similar results. For temperatures results are very similar and the main differences (in favor of CNN) are obtained again for Escandinavia.

[Figure]

*Differences in de-seasonalized correlation for precipitation (left) and temperature (right) found for CNN1 and GLM4 (the latter is taken as reference).*

*These results have been mentioned in the results section of the revised manuscript..*

7. The DOI at line 72 does not work. There is a typo in the first panel of Figure 1, in the caption title.

**Response:** We have updated the version and the correct DOI appears in the revised version of the manuscript.

**Responses to the Reviewers:**

**Configuration and Intercomparison of Deep Learning Neural Models for Statistical Downscaling by J. Baño-Medina, R. Manzanas, and J.M. Gutiérrez. GMD Discussion, https://doi.org/10.5194/gmd-2019-278**

**Reviewer 2**

MAJOR COMMENTS:

1. It would be good to say something more specifically about European applications and the related data needs. Climate change studies are mentioned in the text. But nothing is said about the types of applications/users in the Introduction –and this influences the types of information required – e.g., whether spatial consistency is important, the types of extremes that are relevant.

**Response:** A comment on this has been included in the introduction of the revised manuscript (first paragraph).

2. Line 33-39: This study focuses on the deep learning techniques in the context of perfect prognosis SD. However, it's not clear what the difference between classical SD methods and machine learning techniques is. This needs to be mentioned in the introduction.

**Response:** By classical statistical downscaling techniques we meant the standard well established approaches adopted by the climate community, including generalized linear models and analogs. In the revised manuscript we dropped the term "classical" and used "standard techniques" instead. See, e.g. the introduction (third *paragraph) of the revised manuscript.*

> *"A number of standard perfect prognosis SD (hereafter just SD) techniques have been developed during the last two decades building mainly on (generalized) linear regression and analog techniques (Gutiérrez et al., 2018). These standard approaches are widely used by the downscaling community and several intercomparison studies have been conducted to understand their advantages and limitations taking into account a number of aspects such as temporal structure, extremes, or spatial consistency. In this regard, the VALUE (Maraun et al., 2015) initiative proposed an experimental validation framework for downscaling methods and conducted a comprehensive intercomparison study over Europe with over 50 contributing standard techniques (Gutiérrez et al., 2018).".*

In the machine learning paradigm there are more sophisticated techniques such as random forests, neural networks and support vector machines (among others). We have introduced a clarification concerning this matter in the new version of the manuscript.

3. A warm validation period is selected as surrogate of possible future climate conditions to investigate the suitability of CNN in climate change studies. It should be

better to clearly state how the models produce extremes which are larger than those in the calibration data, and the ability of models to account for changes in the statistics in the future (related to the stationarity assumption)

**Response:** The choice of a warm test period was done in order to perform a preliminary analysis of the extrapolation capabilities of the statistical models. Figures 4 and 6 in the manuscript suggest that the models intercompared are able to work under unseen conditions during the training phase since the local variability was well reproduced according to the metrics evaluated, especially with the convolutional models (with overall unbiased statistics well reproducing the warmer mean value of the test period). We are currently testing the suitability of deep learning approaches to downscale future climate scenarios provided by Global Circulation Models (GCM) and will perform a more detailed analysis related to the stationarity assumption in a future paper. Note that such an analysis is out of the scope of the present paper.

However, in order to address the referee's concern about the reproducibility of extremes larger than those observed in the training data, we computed the 99[th] percentile of temperature in the training period as a robust reference of extreme value, and calculated the frequency of exceeding this value in the test period, for the observations and the GLM1, GLM4 and CNN10 methods. The results are shown in the revised manuscript (Section 4.1, last paragraph).

> *"… In order to further explore this issue, we have also analyzed the capability of the models to produce extremes which are larger than those in the calibration data. To this aim we have considered the 99th percentile over the historical period as a robust reference of extreme value, and calculated the frequency of exceeding this value in the test period for the observations and the GLM1, GLM4 and CNN10 downscaled predictions. The results are shown in Figure 6 and indicate that the three models (in particular the latter two) are able to reproduce the same frequency and spatial pattern of out-of-sample days observed in the test period."*

[Figure]

**Figure 6.** Frequency of exceeding the 99th percentile value of the training period in each of the gridboxes for the observations in the test period and the test predictions of the GLM1, GLM4 and CNN10 models (in columns). Note that a frequency of 1% (in boldface) would indicate the same amount of values exceeding the (extreme) threshold than in the training period.

**4.** Different network architectures of CNN have been evaluated and intercompared in this study. However, the authors should provide more interpretations on the impact of these configuration on model performance. There are a few examples where this is currently done (e.g., lines 215-218, 238-244) but this needs to be done more systematically, and highlighted in the conclusion section.

**Response:** In this study we propose different deep models of increasing levels of complexity and describe their potential added value in Sec. 3; we have included a new paragraph (thirds paragraph) to describe screening process followed to configure the different models. We have also For instance, we consider a convolutional model with linear activation functions (CNN-LM in the manuscript) and its equivalent with nonlinear activation functions (CNN1). This allows to analyze the influence of nonlinearity in the downscaling. Our results show that the introduction of nonlinearities in the model is relevant for precipitation but not for temperature. CNNdense and CNN-PR (see Table 2 of the manuscript for the details) were included in the study since this type of networks are often used in computer vision applications and we wanted to test their potential suitability for statistical downscaling purposes. Whereas CNNdense results from the idea of mixing the spatial patterns learn by the convolutions in the last hidden layers, CNN-RPR is based on the idea that more filter maps are needed as we go further in the net, given the increase of nonlinearity.

A description of the deep models proposed and why they are considered in the current paper can be found in Section 3 of the manuscript. All deep learning models have been intercompared in Figures 4 and 6 in terms of all the proposed metrics for temperature (RMSE, Pearson correlation, bias of the mean, percentile 2 and percentile 98, ratio of standard deviations and spells) and for precipitation (ROCSS, RMSE, Spearman correlation, bias for the mean and percentile 98 and spells).

5. The skill of the various downscaling methods is assessed mostly on spatial variability. How could the CNN reproduce the temporal variability of the local climate? You may want to validate the ability of CNN to represent dry/wet spells and interannual variation.

**Response:** Apart from the correlation already shown in Figures 4 and 6 of the original manuscript, we have included new panels in the revised manuscript in order to address this comment. These figures show now three temporal validation metrics for each target variable: temperature and precipitation. For temperature we show the autocorrelation lag-1 and the bias in the length of the longest warm and cold annual spells. For precipitation we show the relative amplitude of the annual cycle and the relative bias in the length of the longest dry and wet annual spells. These validation indices are described in Sec. 2.2 and Table 1 (the new indices are displayed in the figure below; see Table 1 of the revised manuscript) and results are reported in Sec. 3. In particular, it is shown that no method clearly outperforms the others in terms of spell reproduction, both for temperature and precipitation. Despite there is some spatial variability (spread of the boxplots) the median results are nearly unbiased in all cases.

| Bias (warm annual max spell, WAMSl) | temp. | days |
| Bias (cold annual max spell, CAMS) | temp. | days |
| Bias (wet annual max spell, WetAMS) | precip. | days |
| Bias (dry annual max spell, DryAMS) | precip. | days |
| Bias (lag 1 autocorrelation, AC1) | temp. | - |
| Bias (relative amplitude of the annual cycle) | precip. | - |

**Table 1.** Subset of VALUE metrics used in this study to validate the different downscaling methods considered (see Table 2). The symbol '-' denotes adimensionality.

It is worth to mention that any of the methods considered in this work is specifically designed to reproduce advanced temporal aspects such as spells. In the coming future, we plan to explore other battery of methods which explicitly aim to accurately reproduce the observed temporal structure.

MINOR COMMENTS:
1. Line 13: What does 'classic ones' refer to? Need to make them clear.
2. Line 79: 'such'→'such as'
3. Line 111: 'vale' should be 'value'.
4. Figure 2: The label 'bias' is misleading here, since the map shows the differences between the test and train periods based on observations.

**Response:** We have addressed the minor comments 1, 2, 3 and 4 indicated by the reviewer in the revised manuscript.

5. Figure 4 & 6: The best method is in fact different for each metric, but the same best method (CNN10 for temperature and CNN1 for precipitation) for all metrics is indicated in the figure. How do you choose the best performing method, may be based on one metric?

**Response:** Figures 4 and 6 show the validation results obtained for temperature and precipitation, respectively. For temperature, the CNN10 is the best method according to the RMSE and the de-seasonalized Pearson correlation while keeping unbiased predictions for the mean, and percentiles 2th and 98th. A similar situation occurs for the CNN1 model for precipitation, for which this method outperforms the others in terms of ROCSS, RMSE and Spearman correlation while getting good results for the rest of metrics. For these reasons we chose the CNN10 and CNN1 models to be the 'best' for temperature and precipitation, respectively.

6. Figure 6: Please explain 'DET'(e) and 'STO'(f).

**Response:** 'DET' refers to deterministic and 'STO' to stochastic. We have clarified it in the revised version of the manuscript.

7. Traditional statistical downscaling methods generally require high-resolution observations for model training, thus it is difficult to provide downscaled climate simulations for the regions with little observation data. Is the skill of CNN sensitive to the resolution of observations?

**Response:** To date we have only used deep learning to downscale to resolutions of 0.5º and, despite the sensitivity of downscaling to the observational reference considered is a relevant topic of study, it is out of the scope of this paper. However, we hypothesize that the sensitivity of the downscaling to the predictand's resolution is mainly related to the explicability of the local scale by the predictor's domain rather than by the downscaling method itself (e.g., large-scale predictors have limited predictive skill for convective precipitation and therefore the ability to establish a robust link between both is independent of the statistical method of choice). The benefits of convolutional approaches, such as the ability to treat high-dimensional domains without previous feature selection techniques and the ability to extract non-linear patterns from data, are intrinsic to the multisite-convolutional nature and therefore the

latter skills are expected to be preserved indistinctly of the predictand's resolution. In fact, downscaling to higher resolutions may require a higher degree of nonlinearity and therefore, the skill of deep learning could be even increased in comparison with classical approaches.

In the case of regions with scarce observations, computer vision applications have benefited from a concept called "transfer learning". The idea behind transfer learning is that hidden features learned in a particular task *A* are useful in a similar task *B* and therefore, the trained network *A* (or the first hidden layers) can be used to predict task *B*. In the case of downscaling, though this has not yet been tested to our knowledge, a net trained over a well observed region (e.g., Europe) could be transferable as a pretrained-net over similar areas with less observations available. Though there are still questions to be answered in this topic such as whether the hidden features learned to downscale temperature over Europe (even the most simple ones located in the first hidden layers) would be helpful to downscale in regions with scarce observations which can present their own climatic particularities.

[revised manuscript text omitted]